# DPA-SGG: Dual Prompt Learning with Pseudo-Visual Augmentation for Open-Vocabulary Scene Graph Generation

## Abstract

Open Vocabulary Scene Graph Generation (OVSGG) aims to recognize previously unseen relationships between objects in images, which is essential for robust visual understanding in dynamic real-world scenarios. Recent methods leverage prompt tuning to transfer the rich visual–semantic knowledge of pretrained Vision-Language Models (VLMs), thereby enhancing the recognition ability of unseen predicates. Typically, these methods rely solely on subject and object bounding boxes from seen relationships to extract visual features for guiding visual–semantic alignment during prompt learning. However, this paradigm may lead to two major limitation: 1) **Contextual Blindness**, which means models may overlook broader contextual cues by focusing only on object regions while excluding union regions, making it difficult to distinguish triplets that are visually similar but semantically distinct; 2) **Limited Visual Generalization**, which means models may struggle to transfer effectively to unseen predicates since the training is only restricted to annotated visual regions. To address these limitations, we propose a novel OVSGG framework, termed **DPA-SGG**, consisting of two key components: **Dual Prompt Learning (DLP)**, which introduces two complementary prompts to jointly capture localized object cues and global scene context to better distinguish visually similar relationships; and **Pseudo-Visual Augmentation (PVA)**, which enriches visual diversity by generating a corpus of textual scenes in place of costly visual annotations. Extensive experiments and ablation studies demonstrate the effectiveness of the proposed framework.

## 1 Introduction

Scene Graph Generation (SGG) (Xu et al., 2017; Rotondi et al., 2025), a fundamental scene understanding task, aims to parse an image into a structured semantic representation, typically as a set of visual relation triplets in the form of `<subject, predicate, object>`. Despite being powerful, traditional SGG approaches are limited to a predefined set of object and relationship categories. Open Vocabulary Scene Graph Generation (OVSGG) (He et al., 2022; Yu et al., 2023; Li et al., 2023b) emerges to identify *unseen* relationships between pairwise objects, better suited for the dynamic real-world applications.

Leveraging the advancement in Vision-Language Models (VLMs) (Radford et al., 2021; Li et al., 2022a), existing OVSGG methods typically compare similarity between visual embeddings of the subject and object regions and text embeddings of the class-contained prompts (*e.g.*, "a photo of [relation class]") to achieve OV capability (He et al., 2022; Yu et al., 2023). However, such a set of fixed, context-agnostic text prompts struggles to grasp the rich visual information that defines the specific semantics of a scene. To address this, some works (Li et al., 2023b; Lei et al., 2024; Chen et al., 2024) leverage Large Language Models (LLMs) to generate more discriminative descriptions among relationships. Generally, these methods can be divided into: 1) **Part-level Description**, which decomposes relation detection into several separate components (*e.g.*, subject, object, and spatial) (Li et al., 2023b; Lei et al., 2024), and then leverages LLMs to generate detailed and informative descriptions for each component. 2) **Scene-level Description**, which prompts LLMs to play different roles (*e.g.*, biologist and engineer) to generate comprehensive and diverse descriptions oriented to the scene from different views (Chen et al., 2024).

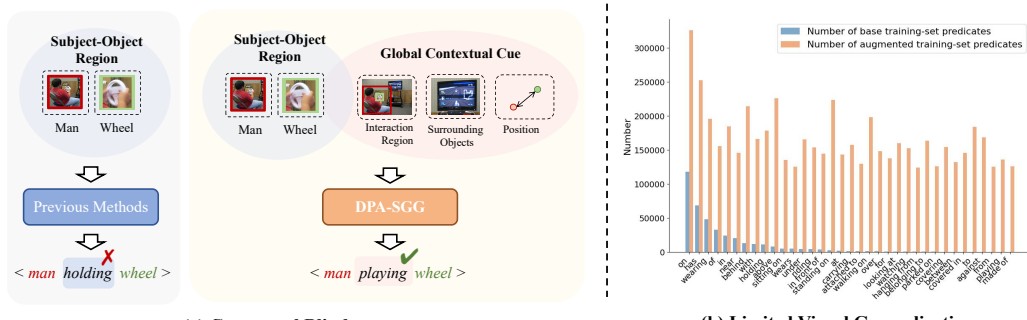

(a) Contextual Blindness       (b) Limited Visual Generalization

Figure 1: The limitation of methods typically relies on subject and object regions. (a) **Contextual Blindness**: Locally similar relation predictions "holding" is mistakenly identified as "playing". (b) **Limited Visual Generalization**: **Blue** bars show the sparse predicate distribution in the original VG dataset, and **orange** bars represent the distribution after pseudo-visual augmentation.

Despite considerable progress, existing OVSGG methods typically rely on subject and object regions to achieve OVSGG capability. However, the current paradigm suffers from the following two limitations: 1) **Contextual Blindness**: Due to the computational bottleneck of union box (the bounding box encompassing both subject and object) clipping, current methods (Li et al., 2023b; Chen et al., 2024; Gao et al., 2023; Menon & Vondrick, 2023) extract solely visual features of the subject and object bounding boxes. Thereby, these models cannot utilize rich context cues, *e.g.*, surrounding objects and the broader interaction region. This narrow focus makes it difficult to disambiguate triplets that are locally similar but semantically distinct. As shown in Figure 1(a), models with a limited focus on localized visual features struggle to distinguish between "man holding wheel" and "man playing wheel", as both actions exhibit minimal visual differences. However, the presence of a television in the background of the whole image provides strong evidence that the predicate is "playing" rather than "holding". 2) **Limited Visual Generalization**. The training of class-contained prompts typically relies on cross-modal alignment with visual features of corresponding image regions (Radford et al., 2021; He et al., 2023; Yu et al., 2023). However, current methods are restricted to annotated visual data, which serves as the sole source of visual knowledge. As shown in Figure 1(b), most predicates in SGG datasets suffer from very limited visual annotations, making it challenging to effectively transfer visual-semantic knowledge to these rare categories. Moreover, relying only on such in-domain annotations inevitably constrains the model's capacity to generalize to unseen predicates in open-vocabulary scenarios. Since annotating visual regions for SGG is highly labor-intensive (Teng & Wang, 2022; Li et al., 2023a), it is therefore desirable to devise a more efficient strategy to augment such visual data.

To this end, we propose **DPA-SGG**, an OVSGG framework that leverages **D**ual **P**rompt learning with pseudo-visual **A**ugmentation. Specifically, the **Dual Prompt Learning (DPL)** is designed to resolve *contextual blindness* via two complementary prompts: a *local prompt* focuses on the fine-grained visual evidence within the specific subject-object region, while a *global prompt* concurrently analyzes the entire image to capture the panoramic scene context. In contrast to the high overhead of union box clipping, our global prompt efficiently integrates a holistic scene context by operating directly on the full image features, enabling our model to distinguish among visually similar relationships. To alleviate the *limited visual generalization* and the high cost of visual annotation, we devise a **Pseudo-Visual Augmentation (PVA)** strategy. This strategy leverages the generative power of LLMs to create a diverse corpus of textual scene descriptions, specifically targeting rare predicates. From this generated text, we extract a rich set of relational triplets. The key insight lies in capitalizing on the tightly aligned embedding space of VLMs. Within this space, the textual embedding of a triplet can serve as a high-fidelity proxy for the visual features of its corresponding scene. As shown in Figure 1(b), our augmentation strategy substantially enriches the predicate distribution with large-scale "pseudo-visual" data. By fine-tuning the model on these augmented datas, DPA-SGG effectively enhances its semantic understanding of rare relationships, building a more robust and generalizable model without requiring any additional annotated images.

To verify the effectiveness of our DPA-SGG, we conduct extensive experiments and ablation studies on the widely used benchmark, Visual Genome (VG) (Krishna et al., 2017). Experimental results show that DPA-SGG outperforms existing OVSGG methods by a large margin.

In summary, we made three main contributions in this paper. **i**) We identify two weaknesses in current OVSGG methods: contextual blindness stemming from reliance on isolated subject and object visual regions, and limited visual generalization for rare visual triplets. **ii**) We propose the DPA-SGG framework that introduces dual prompt learning to efficiently integrate global context and pseudo-visual augmentation to enrich data diversity in a labor-efficient manner. **iii**) Extensive experiments and ablation analysis on the VG benchmark validate our approach, which establishes a new state-of-the-art by a significant margin.

## 2 RELATED WORK

**Scene Graph Generation.** Scene Graph Generation (SGG) has attracted increasing attention as a fundamental task for structured visual understanding, aiming to detect objects in an image and predict their pairwise relationships. Early studies primarily emphasized modeling high-quality visual context (Tang et al., 2019), incorporating techniques such as graph neural networks (Yang et al., 2018) and linguistic priors (Zellers et al., 2018) to refine relational reasoning. However, subsequent research observed that SGG models often suffer from the long-tailed distribution of predicates in prevalent datasets, leading to poor recognition of rare annotated predicates. To address this issue, a variety of strategies have been explored, including feature augmentation (Li et al., 2023a) and label knowledge distillation (Li et al., 2023c). Moreover, since annotating SGG datasets requires excessive human labeling efforts, recent works begin to focus on more economical approaches, such as weakly-supervised learning (Li et al., 2022b) and few-shot learning (Li et al., 2024). By leveraging powerful VLMs, these methods can effectively transfer the rich knowledge of visual-language alignment into SGG models, reducing the reliance on expensive manual annotations.

**Open Vocabulary Learning.** Traditional visual recognition models are typically trained under a closed-set assumption, where they can only recognize predefined categories during training, limiting their adaptability to open-world scenarios. Early efforts addressed this issue through zero-shot learning, which typically leveraged semantic embedding spaces of words (*e.g.*, Word2Vec (Goldberg & Levy, 2014), GloVe (Pennington et al., 2014)) to bridge the gap between seen and unseen categories. However, the limited representational power of these embeddings constrained their scalability. The development of large-scale VLMs, such as CLIP (Radford et al., 2021; Li et al., 2022a), has significantly advanced this area by offering powerful cross-modal representations that support robust knowledge transfer from language to vision. These models have achieved remarkable success in open-vocabulary tasks including image segmentation (Qin et al., 2023) and object detection (Minderer et al., 2022). Building on these advances, recent studies (Chen et al., 2024; Li et al., 2023b) have introduced open-vocabulary paradigms into the SGG task, enabling models to recognize unseen visual relationships and thereby enhancing their scalability to real-world scenarios.

**Foundation Models.** In recent years, large-scale VLMs have emerged as a powerful paradigm for learning cross-modal representations from massive image–text datasets. Representative models such as CLIP (Radford et al., 2021) and ALIGN (Jia et al., 2021) leverage contrastive learning to align visual and textual embeddings within a shared semantic space, enabling robust zero-shot transfer across a wide range of downstream tasks (Li et al., 2023b; Chen et al., 2024). A key factor behind the success of VLMs is their ability to match images with corresponding descriptions while distinguishing mismatched pairs, which enhances their cross-modal understanding. To further adapt VLMs to specific tasks, prompt learning has emerged as a flexible mechanism, providing context or guidance on how the model should apply its knowledge. Beyond hand-crafted prompts or learnable prompts, recent work (Menon & Vondrick, 2022) explores using Large Language Models (LLMs) to automatically generate rich, detailed prompts as inputs to the VLM text encoder. This combination of VLMs and LLMs has shown effectiveness across numerous domains.

## 3 METHODOLOGY

**Formulation.** Given an image $I$, SGG aims to transform it into a structured representation, $\mathcal{G} = \{(s, r, o)|s, o \in \mathcal{O}, r \in \mathcal{R}\}$, where $\mathcal{O}$ represents the set of object categories with bounding boxes and $\mathcal{R}$ denotes the set of predicate categories that describe pairwise relationships between objects. Following the prior works (Li et al., 2023b; Chen et al., 2024), we also focus on the predicate classification task, which predicts the predicate category $r \in \mathcal{R}$ for a given pair of objects $(s, o)$.

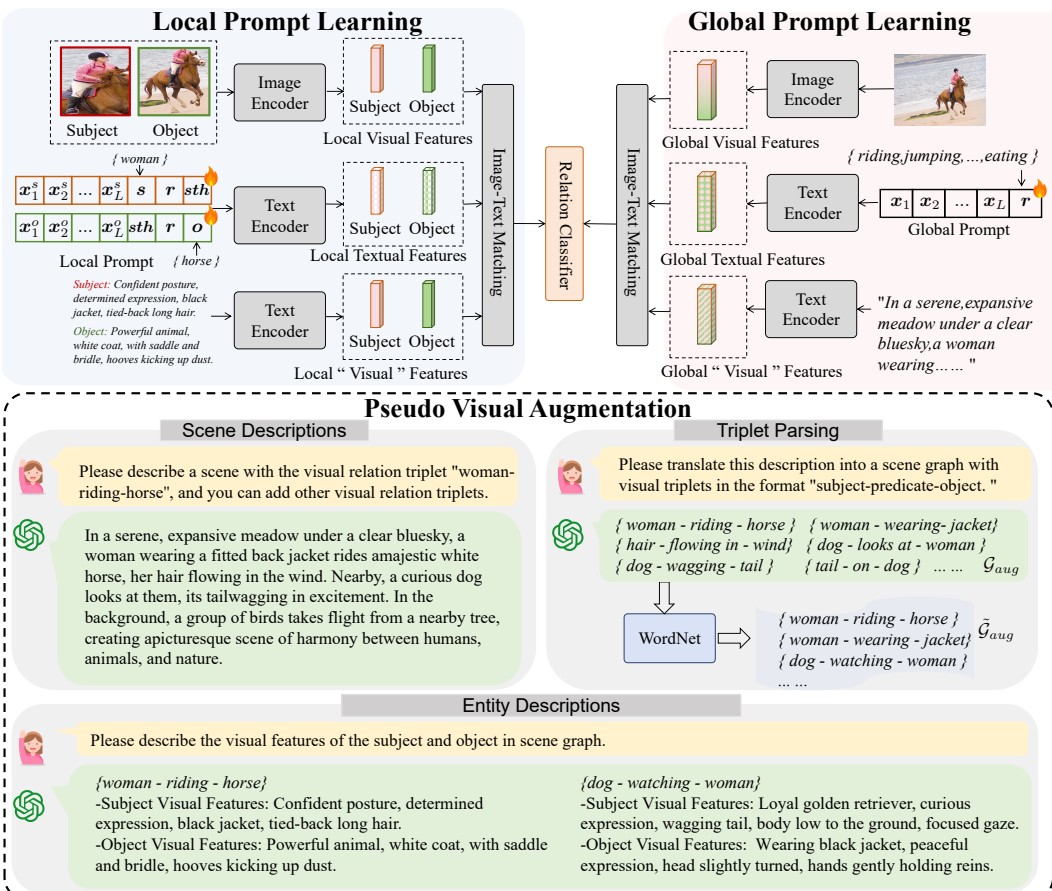

Figure 2: The framework of DPA-SGG. 1) **Local Prompt Learning**: extracts fine-grained features for precise relation prediction. 2) **Global Prompt Learning**: captures coarse-grained features for holistic contextual understanding. 3) **Pseudo Visual Augmentation**: generates scene descriptions and entity descriptions that augment global and local visual features respectively.

Our research addresses the challenge of extending SGG from a traditional closed-set setting to an open-vocabulary paradigm. This transition enables models to recognize previously unseen predicate categories (*i.e.*, novel split) by leveraging knowledge learned from a limited set of observed predicates (*i.e.*, base split) during training.

**Baseline for OVSGG.** Following the standard zero-shot SGG pipeline of previous works (He et al., 2022), a straightforward solution for OVSGG is to obtain visual embeddings for both the subject and object, and then compute the similarity between these visual features and their corresponding text embeddings. The visual embeddings $v_s$ and $v_o$ are typically extracted from the image encoder of a pretrained CLIP (Radford et al., 2021) model, respectively. Similarly, the text embeddings $t_s$ and $t_o$ can be obtained from either a simple class-contained prompt or from some more detailed description-based prompt. The final prediction score for a relation is calculated by summing the cosine similarities between the subject's visual and text embeddings and the object's visual and text embeddings. The relation with the highest score is then selected as the final classification result.

To address the mentioned contextual blindness and limited visual generalization (§1) overlooked by prior works, we propose a novel framework, termed as **DPA-SGG**, for OVSGG. As illustrated in Figure 2, our method comprises two key components: **Dual Prompt Learning** (**DPL**, §3.1) and **Pseudo-Visual Augmentation** (**PVA**, §3.2). These two components work synergistically to enable a context-aware and strong generalization OVSGG.

## 3.1 DUAL PROMPT LEARNING

Compared with prior context-unaware methods, our DPL module introduces a global-local prompt learning framework to enhance contextual perception of models. Specifically, DPL consists of

two branches: *global prompt learning* (§3.1.1) for coarse-grained image-level prediction, and *local prompt learning* (§3.1.2) for fine-grained triplet-level prediction. The final predicate prediction incorporates both the high-level contextual cues from the global prompt and the specific evidence from the local prompt, yielding a contextually grounded and more accurate classification.

### 3.1.1 GLOBAL PROMPT LEARNING

This branch is designed to capture context-aware, panoramic scene information. Rather than relying on the computationally expensive union box cropping strategy, we efficiently leverage the global representation of the entire image to provide panoramic contextual cues. Specifically, we introduce a learnable soft global prompt whose text embedding is aligned with the global visual features of the entire image, enabling the model to adaptively extract richer contextual information and thereby recognize *feasible relations* within the given scene. Inspired by (Zhou et al., 2022; He et al., 2022), we replace hand-crafted text prompts with a set of learnable context vectors that enable dynamically adapting the text embeddings for our OVSGG task, formulated as:

$$\boldsymbol{P}_r^{global} = [\boldsymbol{x}_1, \boldsymbol{x}_2, ..., \boldsymbol{x}_L, \boldsymbol{r}], \tag{1}$$

where $[\boldsymbol{x}_1, \boldsymbol{x}_2, ..., \boldsymbol{x}_L]$ is the prefix $L$ context vectors for global prompt, $\boldsymbol{r}$ denotes the class token embedding vector of relation category $r$. This prompt set is then fed into the CLIP text encoder $En_t(\cdot)$ to obtain the global text embedding as follows:

$$\boldsymbol{t}_r^{global} = En_t(\boldsymbol{P}_r^{global}). \tag{2}$$

Then, we calculate the prediction score of the relation $r_i$ for the global prompt learning branch:

$$p_{r_i}^{global} = \frac{\exp(\phi(\boldsymbol{v}_r^{global}, \boldsymbol{t}_{r_i}^{global})/\tau)}{\sum_{k=1}^{|\mathcal{R}|} \exp(\phi(\boldsymbol{v}_r^{global}, \boldsymbol{t}_{r_k}^{global})/\tau)}, \tag{3}$$

where $\boldsymbol{v}_r^{global}$ is the global visual embeddings of the whole image $I$, extracted by the image encoder $En_v(\cdot)$ of the CLIP, $\phi(\cdot, \cdot)$ represents the cosine similarity between visual embeddings and text embeddings, and $\tau$ is a temperature parameter.

### 3.1.2 LOCAL PROMPT LEARNING

This branch is designed to capture fine-grained details within the specific subject–object region (Li et al., 2023b; Chen et al., 2024). Following a similar strategy as in the global branch, we utilize two learnable soft prompts, one for the subject and the other for the object, formulated as:

$$\begin{aligned} \boldsymbol{P}_s^{loacl} &= [\boldsymbol{x}_1^s, \boldsymbol{x}_2^s, ..., \boldsymbol{x}_L^s, \boldsymbol{s}, \boldsymbol{r}, \boldsymbol{sth}], \\ \boldsymbol{P}_o^{local} &= [\boldsymbol{x}_1^o, \boldsymbol{x}_2^o, ..., \boldsymbol{x}_L^o, \boldsymbol{sth}, \boldsymbol{r}, \boldsymbol{o}], \end{aligned} \tag{4}$$

where $[\boldsymbol{x}_1^s, \boldsymbol{x}_2^s, ..., \boldsymbol{x}_L^s]$ and $[\boldsymbol{x}_1^o, \boldsymbol{x}_2^o, ..., \boldsymbol{x}_L^o]$ represent the learnable local prefix context vectors for the subject and object prompts, respectively, $\boldsymbol{s}$ and $\boldsymbol{o}$ are the class token embedding vectors of subject and object, and $\boldsymbol{sth}$ is the token embedding vector of the word "something". Different from the global branch, the final local prediction score is calculated by summing the cosine similarities of text embedding and their corresponding visual features for both subjects and objects:

$$p_{r_i}^{local} = \frac{\exp(\phi(\boldsymbol{v}_s^{local}, \boldsymbol{t}_{s_i}^{local})/\tau)}{\sum_{k=1}^{|\mathcal{R}|} \exp(\phi(\boldsymbol{v}_s^{loacl}, \boldsymbol{t}_{s_k}^{local})/\tau)} + \frac{\exp(\phi(\boldsymbol{v}_o^{local}, \boldsymbol{t}_{o_i}^{local})/\tau)}{\sum_{k=1}^{|\mathcal{R}|} \exp(\phi(\boldsymbol{v}_o^{loacl}, \boldsymbol{t}_{o_k}^{local})/\tau)}, \tag{5}$$

where $\boldsymbol{v}_s^{local}$ and $\boldsymbol{v}_o^{local}$ are the local visual features of the cropped subject and object bounding boxes, extracted by the image encoder $En_v(\cdot)$ of the CLIP.

**Relation Classification.** To ensure predictions are consistent with both fine-grained details and the broader scene context, we fuse the local and global scores via a geometric mean. The key advantage of this multiplicative fusion is its ability to suppress unfeasible predictions: if either the global or local context assigns a probability of zero to a predicate, the fused prediction score will also be zero, effectively eliminating that predicate from consideration. The fusion process can be formulated as:

$$p_{r_i} = (p_{r_i}^{local})^\lambda \cdot (p_{r_i}^{global})^{1-\lambda}, \tag{6}$$

where the hyperparameter $\lambda$ controls the influence of the local versus the global score. The final predicted relation is the class with the highest fused score.

## 3.2 Pseudo-Visual Augmentation

This module aims to address the challenge of limited visual generalization caused by the lack of annotated visual data. Considering the high cost and limited availability of annotated visual data, we leverage the fact that *visual and text encoders of a pretrained VLM operate within a tightly aligned semantic space*. This allows us to generate a diverse corpus of textual descriptions as "pseudo-visual" data to replace the real images (Guo et al., 2023).

Specifically, we generate *scene descriptions* and *entity descriptions* to respectively replace the global visual features of the entire image and the local visual features of the subject and object regions.

**Scene Descriptions.** We introduce a scene description generation prompt to make LLMs generate comprehensive and panoramic scene descriptions $\mathcal{D}_{scene}$ for each given visual relationship triplet $(s, r, o) \in \mathcal{G}$. The generation process of $\mathcal{D}_{scene}$ can be expressed as:

$$\mathcal{D}_{scene} = \text{LLM}(\underbrace{\text{in-context examples}, (s, r, o), \text{instruction}}_{\text{prompt input}}), \tag{7}$$

where $\text{LLM}(\cdot)$ is the decoder of the LLMs, *in-context examples* provide some examples of the desired generation results to make the LLM generate analogous results, $(s, r, o)$ is the specific triplet class to be included in the scene description. The *instruction* is the sentence used to command the LLM to generate the description, *e.g.*, "Please describe a scene with the visual relation 'person-riding-horse', and you can add other visual relation triplets."

Subsequently, the global prompt learning branch is further trained based on pseudo-visual data generated from scene descriptions. The prediction score can calculated as:

$$\tilde{p}_{r_i}^{global} = \frac{\exp(\phi(\tilde{\boldsymbol{v}}_r^{global}, \boldsymbol{t}_{r_i}^{global})/\tau)}{\sum_{k=1}^{|\mathcal{R}|} \exp(\phi(\tilde{\boldsymbol{v}}_r^{global}, \boldsymbol{t}_{r_k}^{global})/\tau)}, \tag{8}$$

where $\tilde{\boldsymbol{v}}_r^{global} = En_t(\mathcal{D}_{scene})$ is the global "visual" embeddings of the generated scene descriptions, extracted by the text encoder $En_t(\cdot)$ of the CLIP.

To mitigate the severe learning bias induced by the long-tail distribution (Tang et al., 2020) of SGG datasets, we adopt a dynamic generation strategy where the number of generated triplets is set inversely proportional to their frequency in the original dataset. Formally, it is defined as:

$$N_{gen}(s, r, o) = \frac{\gamma}{f(s, r, o)}, \tag{9}$$

where $N_{gen}$ is the number of new generated descriptions for a given triplet $(s, r, o)$, $f(s, r, o)$ is its frequency in the original dataset, and $\gamma$ is a scaling hyperparameter.

**Triplet Parsing.** Noting that the generated scene description contains not only the given input triplet but also a variety of other possibly co-occurring triplets, we can parse them from the scene description to obtain extra trplet samples. With the help of LLMs, these triplets can be extracted effectively. However, since the elements (*i.e.*, subject, predicate, object) in the extracted triplets may not always correspond directly to the predefined categories in the dataset, an alignment operation is necessary to map them to an appropriate existing categories before they can be utilized for model training. The whole parsing process consists of two steps: 1) **Extraction**: We design a triplet extraction prompt to extract all meaningful relationship triplets from scene descriptions as follows:

$$\mathcal{G}_{aug} = \text{LLM}(\underbrace{\text{in-context examples}, \mathcal{D}_{scene}, \text{instruction}}_{\text{prompt input}}). \tag{10}$$

Similarly, *in-context examples* are analogous examples, the *instruction* sentences are designed to enable LLM to generate all possible relationship triplets in the scene, *e.g.*,"Please translate this description into a scene graph with visual triplets in the format subject-predicate-object". The second step is 2) **Alignment**: For each element $(s_{aug}, o_{aug}, r_{aug})$ in the extracted triplets $\mathcal{G}_{aug}$, we compute its semantic distance from the predefined categories using WordNet (Miller, 1995) and select the closest one as the pre-alignment result. We further introduce a maximum distance threshold $\delta$ to obtain the alignment result. If the minimum semantic distance for any component of the pre-alignment triplet exceeds this threshold, this triplet will be discarded in the final alignment result.

$$\tilde{s}_{aug} = \underset{s \in \mathcal{S}}{\arg\min} \, \text{dist}(s_{aug}, s), \quad \tilde{r}_{aug} = \underset{r \in \mathcal{R}}{\arg\min} \, \text{dist}(r_{aug}, r), \quad \tilde{o}_{aug} = \underset{o \in \mathcal{O}}{\arg\min} \, \text{dist}(o_{aug}, o), \tag{11}$$

where $\text{dist}(\cdot, \cdot)$ is a semantic distance function Then we can obtain the final parsed triplets $\tilde{\mathcal{G}}_{aug} = \{(\tilde{s}_{aug}, \tilde{r}_{aug}, \tilde{o}_{aug}) \mid \text{dist}(s_{aug}, \tilde{s}_{aug}) < \delta \ \& \ \text{dist}(r_{aug}, \tilde{r}_{aug}) < \delta \ \& \ \text{dist}(o_{aug}, \tilde{o}_{aug}) < \delta\}$ in the scene description, which serves as training targets for prompt learning.

**Entity Descriptions.** To generate entity descriptions that can serve as substitutes for local visual features, we adopt a scene-specific entity (*i.e.*, subject and object) description generation prompt, guiding LLM to produce descriptions $D_{entity}$ that reflect the fine-grained context of each scene for each entity class:

$$\mathcal{D}_{entity} = \text{LLM}(\underbrace{\text{in-context examples}, \mathcal{D}_{scene}, (s, o, r), \text{instruction}}_{\text{prompt input}}). \tag{12}$$

We utilize *instruction* like "Please describe the visual features of the subject and object in scene graph". Then the prediction score can be calculated as:

$$\tilde{p}_{r_i}^{local} = \frac{\exp(\phi(\tilde{\boldsymbol{v}}_s^{local}, \boldsymbol{t}_{s_i}^{local})/\tau)}{\sum_{k=1}^{|\mathcal{R}|} \exp(\phi(\tilde{\boldsymbol{v}}_s^{loacl}, \boldsymbol{t}_{s_k}^{local})/\tau)} + \frac{\exp(\phi(\tilde{\boldsymbol{v}}_o^{local}, \boldsymbol{t}_{o_i}^{local})/\tau)}{\sum_{k=1}^{|\mathcal{R}|} \exp(\phi(\tilde{\boldsymbol{v}}_o^{loacl}, \boldsymbol{t}_{o_k}^{local})/\tau)}, \tag{13}$$

where $\tilde{\boldsymbol{v}}_s^{local}$ and $\tilde{\boldsymbol{v}}_o^{local}$ are the local "pseudo-visual" embeddings of the generated subject and object descriptions, extracted by the text encoder $En_t(\cdot)$ of the CLIP.

## 3.3 TRAINING OBJECTIVE

In the training stage, we train each component in DPA-SGG separately, including both global prompt learning and local prompt learning.

**Training Objective of Global Prompt Learning.** To train the global prompt learning branch, we adopt a powerful ranking loss to encourage the model to predict high scores of positive categories and low scores of negative categories. Our training process consists of two stages. In the first stage, the model is trained on the original dataset, which can be defined as:

$$\mathcal{L}_{global} = \frac{1}{|\mathcal{B}|} \sum_{(r_{\text{pos}}, r_{\text{neg}})} \max(1 + p_{r_{\text{neg}}}^{global} - p_{r_{\text{pos}}}^{global}, 0), \tag{14}$$

where $(r_{\text{pos}}, r_{\text{neg}}) \in \mathcal{B}$ is a pair of positive and negative predicate category for each triplet sample in the batch $\mathcal{B}$, $|\mathcal{B}|$ is the number of triplets in the batch, and $p_{r_{\text{pos}}}^{global}$ is the classification score of positive predicate category. During training, the global prompt learns to align the text embeddings of multi-label base categories for each image by minimizing $\mathcal{L}_{global}$. In the second stage, we use the same training strategy on the pseudo-visual data.

**Training Objective of Local Prompt Learning.** Similar to the global prompt training, we also use a two-stage training process, leveraging both visual and pseudo-visual data with a consistent training strategy. This component utilizes a cross-entropy loss, calculated in the first stage as:

$$\mathcal{L}_{local} = -\frac{1}{|\mathcal{B}|} \sum_{\mathcal{B}} \log \frac{\exp(p_{r_{\text{GT}}}^{local})}{\sum_{k=1}^{|\mathcal{R}|} \exp(p_{r_k}^{local})}, \tag{15}$$

where $|\mathcal{B}|$ is the number of triplets in the batch $\mathcal{B}$, $r_{\text{GT}}$ is the ground-truth relation category. In the second stage, we train with the same loss function on the pseudo-visual data generated by our pseudo-visual augmentation module.

**Total Loss.** The total training objective is the sum of these two loss:

$$\mathcal{L} = \mathcal{L}_{global} + \mathcal{L}_{local}. \tag{16}$$

## 4 EXPERIMENT

### 4.1 EXPERIMENT SETUP

**Datesets.** We evaluated our method on the challenging and widely-used benchmark VG (Krishna et al., 2017): which consists of 50 predicate classes and 150 object classes. Following previous

Table 1: Quantitative results (§4.2) on VG `base` and `novel`.

| Method | Split | R@20↑ | R@50↑ | R@100↑ | mR@20↑ | mR@50↑ | mR@100↑ |
|---|---|---|---|---|---|---|---|
| CLS[ICML21] | | 2.1 | 3.2 | 3.9 | 7.0 | 9.0 | 10.9 |
| Epic[ICCV23] | base | - | 22.6 | 27.2 | - | - | - |
| SDSGG[NIPS24] | | 18.7 | 26.5 | 31.6 | 9.2 | 12.4 | 14.8 |
| **Ours** | | **42.42** | **53.63** | **59.03** | **10.53** | **15.00** | **17.73** |
| CLS[ICML21] | | 13.2 | 18.1 | 22.2 | 11.5 | 17.9 | 23.8 |
| Epic[ICCV23] | novel | - | 7.4 | 9.7 | - | - | - |
| SDSGG[NIPS24] | | 18.4 | 25.4 | 29.6 | 17.1 | 25.2 | 31.2 |
| **Ours** | | **25.25** | **32.86** | **36.75** | **24.48** | **31.47** | **35.44** |

Table 2: Effectiveness of each component.

| Local | Global | PVA | Split | R@20↑ | R@50↑ | R@100↑ | mR@20↑ | mR@50↑ | mR@100↑ |
|---|---|---|---|---|---|---|---|---|---|
| ✓ | | | | 38.36 | 48.75 | 54.36 | 6.12 | 10.26 | 12.92 |
| ✓ | ✓ | | base | 40.05 | 50.87 | 57.94 | 7.31 | 11.97 | 14.42 |
| ✓ | ✓ | ✓ | | **42.42** | **53.63** | **59.03** | **10.53** | **15.00** | **17.73** |
| ✓ | | | | 21.59 | 26.04 | 28.16 | 21.22 | 26.13 | 28.35 |
| ✓ | ✓ | | novel | 23.05 | 28.13 | 30.96 | 22.82 | 27.45 | 29.87 |
| ✓ | ✓ | ✓ | | **26.53** | **31.77** | **34.70** | **25.94** | **30.78** | **33.58** |

OVSGG work (Yu et al., 2023; Chen et al., 2024), the VG dataset is divided into a base and a novel split. The base split comprises 35 relation categories, which account for 70% of the total categories used for training. The novel split comprises 15 relation categories, which contain the remaining 30% of categories unseen during training.

**Evaluation Metrics.** We evaluated our method on the standard predicate classification (PredCls) task. The evaluation metrics used are Recall@K (R@K) and mean Recall@K (mR@K).

**Baselines.** We compared our proposed method DPA-SGG with three strong baselines: 1) **CLS** (Radford et al., 2021), which only uses the category name as prompts to compute the similarity between image and text. 2) **Epic** (Yu et al., 2023), which introduces an entangled cross-modal prompt and leverages contrastive learning. 3) **SDSGG** (Chen et al., 2024), which leverages scene-specific descriptions as text embedding.

## 4.2 QUANTITATIVE COMPARISON RESULT.

In this work, we evaluated the performance on both the base and novel splits of the VG (Krishna et al., 2017) dataset. As shown in Table 1, we have the following observations: 1) The CLS baseline, which relies on simple class-based prompts, demonstrated inferior performance, particularly on the base split. This is because the base split has a larger number of categories (35 *vs.* 15) for the novel split and prepositions (*i.e.*, "on", "of", and "at") in base split that inherently lack specific visual semantics, making them difficult to distinguish by CLIP. 2) The Epic baseline achieves performance gain on the base split (*i.e.*, 22.6% R@50 and 27.2% R@100), demonstrating the effectiveness of its entangled cross-modal prompt on base data. However, its performance dramatically drops to 7.4% R@50 and 9.7% R@100 on the novel split, indicating a severe overfitting problem. 3) SDSGG achieves a light performance gain on both splits due to its scene-specific prompts strategy. 4) The proposed DPA-SGG exhibits significant performance gains across all metrics compared to all baseline models, *e.g.*, 31.6 → **59.03**% R@100 on base split and 31.2% → **35.44**% mR@100 on novel split. This indicates the effectiveness of DPA-SGG framework in OVSGG.

## 4.3 ABLATION STUDIES.

We conducted a series of ablation studies on VG (Krishna et al., 2017) to thoroughly evaluate our proposed components.

**Key Component Analysis.** We analyzed the influence of three major components for DPA-SGG: 1) **Global**, which denotes the global prompt learning part in GPL (§3.1.1); 2) **Local**, which denotes the local prompt learning part in GPL (§3.1.2); 3) **PVA**, which denotes leverages pseudo-visual augmentation module (§3.2). From the results in Table 2, we have the following conclusions: 1) By adding the global prompt learning to capture the panoramic scene information, it reached a clear

Table 3: Effectiveness of hyper-parameter $\lambda$.

| $\lambda$ | Split | R@20↑ | R@50↑ | R@100↑ | mR@20↑ | mR@50↑ | mR@100↑ |
|---|---|---|---|---|---|---|---|
| 0.0 |  | 26.73 | 37.45 | 43.48 | 5.75 | 9.29 | 11.93 |
| 0.3 |  | 36.89 | 47.58 | 52.86 | 8.13 | 11.76 | 14.28 |
| 0.5 | base | 40.82 | 51.99 | 57.28 | 9.43 | 13.50 | 16.11 |
| 0.8 |  | **42.42** | **53.63** | **59.03** | **10.53** | **15.00** | **17.73** |
| 1.0 |  | 26.73 | 37.45 | 43.47 | 5.75 | 9.29 | 11.93 |
| 0.0 |  | 24.85 | 30.68 | 34.32 | 23.65 | 29.56 | 33.09 |
| 0.3 |  | 25.25 | **32.86** | **36.75** | 24.48 | **31.47** | **35.44** |
| 0.5 | novel | 25.76 | 31.78 | 35.63 | 25.16 | 30.68 | 34.56 |
| 0.8 |  | **26.53** | 31.77 | 34.70 | **25.94** | 30.78 | 33.58 |
| 1.0 |  | 27.12 | 32.07 | 34.77 | 25.64 | 30.77 | 33.33 |

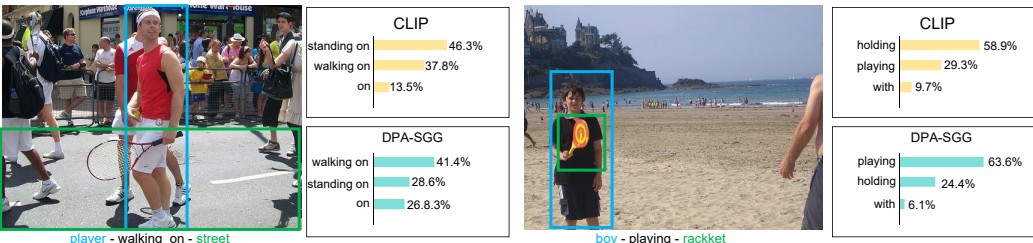

Figure 3: Visual result (§4.4) on VG (Krishna et al., 2017).

performance improvement, *e.g.*, increases from 10.26% to 11.97% mR@50 on the base split, and from 26.13% to 27.45% mR@50 on the novel split; 2) The introduction of the PVA component led to a further performance gain, *e.g.*, 29.85% → **33.58**% mR@100 on novel split; 3) By integrating all the key component, DPA-SGG delivered the best performance across all metrics.

**Analysis of Hyperparameters.** Table 3 shows the results of varying values of $\lambda$ in Eq. 6 (§3.1.2), which controls the influence of the local score relative to the global score. On the base split, our framework achieved its best performance with a $\lambda = 0.8$. This suggests that a higher weight on the fine-grained local scores is beneficial for the base split. However, on the novel split, the highest scores for R@20 and mR@20 are achieved at $\lambda = 0.8$ (26.53% and 25.94%, respectively), but the best scores for R@50, R@100, mR@50, and mR@100 are obtained with $\lambda = 0.3$. This indicates that fine-grained information is highly effective for making high-confidence predictions.

## 4.4 QUALITATIVE COMPARISON RESULT

As depicted in Figure 3, we visualized qualitative comparisons of DPA-SGG against CLIP (Radford et al., 2021), which relies solely on subject and object visual features extracted from their bounding boxes on VG (Krishna et al., 2017). We can observe that CLIP incorrectly predicts an ambiguous relation "standing on", whereas DPA-SGG accurately identifies "walking on". This demonstrated that DPA-SGG is an effective framework for OVSGG, even in challenging scenarios where relations are difficult to disambiguate.

## 5 CONCLUSION

In this paper, we focused on two overlooked limitations of current OVSGG methods: contextual blindness and limited visual generalization. To overcome these issues, we proposed a novel framework, DPA-SGG, which leverages a dual prompt learning strategy to capture both local and global context and a pseudo-visual augmentation module to enhance semantic comprehension of rare relationships, thereby creating a more robust and generalizable model without relying on additional annotated images. Extensive experiments on the VG dataset demonstrated the effectiveness of our proposed DPA-SGG. We believe the introduction of our DPA-SGG framework will not only set a new benchmark for OVSGG but also encourage the community to explore the potential of integrating context-aware and data-efficient paradigms for other vision-language tasks.

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

APPENDIX

This appendix is organized as follows:

- §A elaborates the implementation details of DPA-SGG.
- §B presents the reproducibility satatement.
- §C covers the large language model usage statement.

## A  IMPLIMENTATION DETAILS

For the CLIP model backbone, we utilized ViT- B/32 (Radford et al., 2021) as both the visual and text encoder, setting the dimension $C$ to 512, and we use the default logits scale $\tau$ (Eq. 8 and Eq. 13) with the same pretrained CLIP. Our implementation of DPA-SGG is based on PyTorch and two NVIDIA GTX 3090 GPUs. we used the Adam optimizer (Loshchilov & Hutter, 2017) with a learning rate of 1e-2 and a batch size of 24 as default. In the DPL module, we set prefix length $L = 12$ ( Eq. 1 and Eq. 4), and hyperparameter $\lambda = 0.8$ (Eq. 6) as default. In PVA module, we set the scaling hyperparameter $\gamma = 200$ (Eq. 9), and distance threshold $\delta = 0.5$ (§3.2) .

## B  REPRODUCIBILITY STATEMENT

In the spirit of open science and to facilitate future research, the source code for the DPA-SGG framework will be made publicly available upon acceptance of this paper for full verification and extension. The implementation details of our model architecture, training procedures, and hyperparameters are provided in Appendix §A.

## C  LARGE LANGUAGE MODEL USAGE STATEMENT

We utilized the Gemini-2.5 Pro (Comanici et al., 2025) as an LLM to use as a writing assistant to polish the language of the manuscript. Additionally, we use GPT-3.5-trubo (Floridi & Chiriatti, 2020) as an LLM to generate descriptions in PVA module (§3.2).

