# OpenReview forum: "DPA-SGG: Dual Prompt Learning with Pseudo-Visual Augmentation for Open-Vocabulary Scene Graph Generation"
_ICLR.cc/2026/Conference — ICLR 2026 Conference Withdrawn Submission_

### Official Review · Reviewer_Pfc6 · 2025-10-29

**Soundness:** 2
**Presentation:** 2
**Contribution:** 2
**Rating:** 4
**Confidence:** 4

**Summary:**

This paper targets Open-Vocabulary Scene Graph Generation (OVSGG) and identifies two concrete and under-addressed weaknesses in existing methods: (1) contextual blindness due to over-reliance on subject–object box embeddings without global scene reasoning, and (2) limited visual generalization caused by dependence on scarce annotated visual regions. The proposed framework DPA-SGG introduces (i) Dual Prompt Learning, jointly optimizing a local fine-grained prompt and a global panoramic prompt without costly union box cropping — and (ii) Pseudo-Visual Augmentation, which uses LLM-generated scene and entity descriptions as high-fidelity textual “visual surrogates” to balance long-tail predicate learning.

**Strengths:**

The paper does not simply state “lack of generalization” as a vague complaint; it isolates two failure modes with concrete causal mechanisms and illustrates them clearly (e.g., “holding vs playing” failure tied to missing global context). The proposed solutions are tightly engineered to address these problems: local–global prompt decomposition directly corresponds to contextual blindness, and LLM-driven semantic pseudo-visual data directly corresponds to long-tail underrepresentation. Both components are plausibly novel and computationally efficient. They avoid expensive union box handling or any need to render synthetic images.

**Weaknesses:**

Despite using LLMs to replace visual data, the paper does not rigorously quantify when textual pseudo-visual information begins to diverge from real visual modality, nor does it show failure or sensitivity analysis to hallucinated relationships or domain drift in generated scenes.

The method also relies on semantic alignment via WordNet matching during triplet parsing. This alignment step may silently discard informative but paraphrastic relations, but the effect is not analyzed.

Finally, although computation is lighter than union-box pipelines, the framework still introduces multiple LLM calls in training. The paper does not present a cost or scalability comparison versus alternative augmentation approaches (e.g., diffusion-based synthetic views).

**Questions:**

Since the PVA module replaces real image features with text encoder embeddings only, how do you ensure that the pseudo-visual data truly preserves visual grounding instead of merely amplifying linguistic priors? Do you have any failure or drift analysis?

The triplet alignment step discards extracted relations if their WordNet semantic distance exceeds a threshold. What percentage of generated triplets is actually discarded, and how sensitive is final performance to δ?

You claim that union-box cropping is computationally expensive, could you provide concrete FLOP or latency comparisons between your global branch and a union-box-based baseline?

---

### Official Review · Reviewer_wVCn · 2025-10-31

**Soundness:** 2
**Presentation:** 2
**Contribution:** 1
**Rating:** 4
**Confidence:** 3

**Summary:**

The paper proposes DPA-SGG, a framework for open-vocabulary scene graph generation (OVSGG). It has two main components: (1) Dual Prompt Learning (DPL) with a global soft prompt aligned to whole-image features and a local soft prompt aligned to subject/object crops; and (2) Pseudo-Visual Augmentation (PVA) that uses LLM-generated scene and entity descriptions to create additional ''pseudo-visual'' training signals by feeding those texts into CLIP’s text encoder in lieu of images. The authors claim DPA-SGG alleviates ''contextual blindness'' from using only subject/object regions and improves generalization to rare/unseen predicates, reporting sizable gains on Visual Genome PredCls base/novel splits over three baselines (CLS, Epic, SDSGG) with ablations for each component and a λ fusion hyper-parameter.

**Strengths:**

1. Reasonable high-level goal. Addressing ``contextual blindness'' of subject/object crops by adding image-level context is a valid objective for SGG and OVSGG. The paper’s global–local split and multiplicative fusion are clearly described.
2. Compute-light implementation. The method builds on CLIP with prompt tuning and avoids heavy architectural changes, which could be practical.1
3. Some ablations. A component ablation (Global / Local / PVA) and a $\lambda$-fusion description are provided.

**Weaknesses:**

1. Comparison with the prompt-based OVSGG and dual-prompt literature. The core design (i.e., learnable soft prompts plus LLM-generated descriptions to steer CLIP) largely reuses known patterns in OVSGG/prompt-tuning. Even the paper’s own ``inspired by (Zhou; He) Line 227`` phrasing suggests incremental reuse rather than a new principle. The paper should re-position its novelty beyond "replace hand-crafted prompts with learnable ones + add descriptions."
2. PVA’s modality gap and validity are under-analyzed. PVA replaces images with text embeddings from CLIP’s text encoder as "pseudo-visual" features for both global and local branches, creating a training-testing modality mismatch (text→text at train; image→text at test) that can over-index on language priors. No diagnostics or controls are provided.
3. Ambiguity and potential leakage in global supervision. The global branch uses a ranking loss and says it ``aligns the text embeddings of multi-label base categories for each image``, which risks leaking scene-level predicate presence into unrelated pairs during training. How are image-level positives defined from triplets? Is supervision pair-conditioned?
4. No sufficient evidence for the "efficiency" motivation. The paper claims union-box pipelines are computationally expensive and that the global prompt is efficient, but provides no runtime/FLOPs/latency comparisons.

**Questions:**

1. What happens if you use image-based synthetic data (even crude renders) instead of text-only PVA? Please report comparisons and failure cases.
2. What fraction of LLM-extracted triplets are discarded by $\delta$? Provide alignment precision/recall and sensitivity to $\delta, \gamma$.
3. Please include profiling vs. a union-box baseline and with/without PVA (train & inference).

---

### Official Review · Reviewer_4CHb · 2025-11-02

**Soundness:** 2
**Presentation:** 2
**Contribution:** 2
**Rating:** 2
**Confidence:** 4

**Summary:**

The paper proposes an open-vocabulary visual relation classification approach within the broader context of scene graph generation (SGG). The method introduces two main components to improve relation representation and generalization: (1) a dual prompt learning module that incorporates a learnable scene-level global prompt for contextual cues, and (2) a text-based data augmentation module that synthesizes pseudo samples for tail predicates using large language models. The approach is evaluated on the Visual Genome dataset under the Predicate Classification (PredCLS) setting.

**Strengths:**

- The proposed prompt learning and text-based augmentation seems reasonable for open-vocabulary relation modeling.

- The ablation study shows the contribution of each proposed component.

**Weaknesses:**

- Limited problem scope. Although the paper claims to address Open-Vocabulary Scene Graph Generation (OV-SGG), it only tackles the predicate classification (PredCLS) subtask. This restricted setting overlooks the challenges of object detection and triplet prediction, which are central to OV-SGG. As a result, the work provides an incomplete solution.
- Unconvincing motivation. The paper frames “contextual blindness” as a core issue, but this claim is not well justified, since most SGG pipelines already use union-box features of interacting object pairs to capture context. Moreover, several prior OV-SGG approaches have incorporated pre-trained VLMs (e.g., CLIP-based encoders) to obtain relational features beyond annotated visual data.
- Technical concerns. 1) The global prompt learning relies on fixed pre-trained visual encoders, which limits its ability to capture fine-grained relational cues from weak object regions. 2)  The pseudo-visual augmentation depends on the accuracy and consistency of LLM-generated scene descriptions, yet the robustness of this process is not analyzed or quantified. 3) The training objectives for the two modules are different, and the paper does not present a unified optimization framework, leaving the overall design somewhat ad hoc.
- Insufficient experimental evaluation. 1) The evaluation is limited to PredCLS, while SGDet (Scene Graph Detection) should be included to better assess real-world performance. 2) The benchmark coverage is narrow; results on larger and more diverse datasets such as OpenImages V6 would strengthen the claims. 3) The method lacks comparisons with several recent and relevant OV-SGG approaches, including LLaVA-SpaceSGG (WACV 2024), OvSGTR (ECCV 2024), RAHP (AAAI 2025), and Navigating the Unseen (CVPR 2025)—which are also missing from the related work section.

**Questions:**

- Can the authors justify why only PredCLS was evaluated and how the proposed method would generalize to SGDet or SGCls?

- How robust is the LLM-based pseudo-augmentation to incorrect or inconsistent text generation?

- How does the method perform compared to recent OV-SGG models listed above?

---

### Official Review · Reviewer_DGH4 · 2025-11-03

**Soundness:** 3
**Presentation:** 3
**Contribution:** 3
**Rating:** 8
**Confidence:** 4

**Summary:**

The authors describe a new Open Vocabulary Scene Graph Generation (OVSGG)
method that aims to overcome contextual blindness stemming from reliance on
isolated subject and object visual regions, and limited visual generalization
for rare visual triplets. The method is called DPA-SGG, which includes a Dual
Prompt Learning (DLP) component as well as Pseudo-Visual Augmentation (PVA).
The method shows promising results.

**Strengths:**

S1: The proposed DPA-SGG method shows state-of-the-art results, with
significant improvement over competing methods.

S2: Open Vocabulary Scene Graph Generation (OVSGG) is a difficult task.

**Weaknesses:**

W1: Figure 1 is too small. The thumbnail images and some of the text are very
small. Some increase in size would be helpful.

**Questions:**

Please see W1 under Weaknesses.

---

### Note · Authors · 2025-12-01

I have read and agree with the venue's withdrawal policy on behalf of myself and my co-authors.